# Martini 3 Model of Cellulose Microfibrils: On the Route to Capture Large Conformational Changes of Polysaccharides

**DOI:** 10.3390/molecules27030976

**Published:** 2022-02-01

**Authors:** Rodrigo A. Moreira, Stefan A. L. Weber, Adolfo B. Poma

**Affiliations:** 1Biosystems and Soft Matter Divison, Institute of Fundamental Technological Research, Polish Academy of Sciences, Pawińskiego 5B, 02-106 Warsaw, Poland; rams@ippt.pan.pl; 2Max Planck Institute for Polymer Research, Ackermannweg 10, 55128 Mainz, Germany; webers@mpip-mainz.mpg.de; 3International Center for Research on Innovative Biobased Materials (ICRI-BioM)—International Research Agenda, Lodz University of Technology, Żeromskiego 116, 90-924 Lodz, Poland

**Keywords:** cellulose I allomorphs, cellulose II, Martini 3, large conformational changes, twist, molecular dynamics, coarse-grained model, aggregation

## Abstract

High resolution data from all-atom molecular simulations is used to parameterize a Martini 3 coarse-grained (CG) model of cellulose I allomorphs and cellulose type-II fibrils. In this case, elementary molecules are represented by four effective beads centred in the positions of O2, O3, C6, and O6 atoms in the D-glucose cellulose subunit. Non-bonded interactions between CG beads are tuned according to a low statistical criterion of structural deviation using the Martini 3 type of interactions and are capable of being indistinguishable for all studied cases. To maintain the crystalline structure of each single cellulose chain in the microfibrils, elastic potentials are employed to retain the ribbon-like structure in each chain. We find that our model is capable of describing different fibril-twist angles associated with each type of cellulose fibril in close agreement with atomistic simulation. Furthermore, our CG model poses a very small deviation from the native-like structure, making it appropriate to capture large conformational changes such as those that occur during the self-assembly process. We expect to provide a computational model suitable for several new applications such as cellulose self-assembly in different aqueous solutions and the thermal treatment of fibrils of great importance in bioindustrial applications.

## 1. Introduction

In nature, biomass is a renewable resource, and today it is considered a key material in the circular economy plan. This natural resource is chemically composed of insoluble carbohydrates (e.g., lignocellulose and cellulose) which under the action of several enzymes can be converted into small monomeric subunits and then easily undergo degradation [1]. The extraction and purification at high quality of polysaccharides for industrial applications (mostly due to its inherent biodegradability) [2] are typically carried out by chemical and physical processes, such as contact with ionic liquids [3] or TEMPO-oxidation [4] and ultrasonic separation [5], respectively. These processes aim to destabilize the polar and electrostatic forces between cellulose chains in fibrils.

Molecular Dynamics (MD) simulation is the computational tool capable of investigating the underlying atomistic mechanisms in macromolecular systems at very short length and time scales (e.g., nm and μs). In this regard, the inner hydrophobic structure of cellulose I allomorphs and type-II fibrils has been characterized by MD methods [6,7,8]. As a result, the hydrogen bond (HB) network necessary to build an elementary microfibril has been quantitatively described in the ground state of the microfibrils in solution [7]. Note that a relatively strong presence of the O3–H⋯O5 intrachain and O6–H⋯O3 interchain HBs are relatively established, and it is believed they are responsible for the stability of the layered structures. This result was validated by X-ray and neutron fiber diffraction [9,10] and demonstrated the absence of O–H⋯O interactions between layers and a larger presence of C–H⋯O HBs between cellulose sheets. Thus, it is expected that some C–H⋯O hydrogen bonds and van der Waals forces may contribute to the stability of cellulose I.

In the age of fast computational processing and machine learning (ML), larger and more complex systems in equilibrium can be modeled by advanced all-atom MD and ML tools, such as viral diseases [11,12], material design [13], etc. However, out of equilibrium processes such as large conformational changes of biomolecules, e.g., the unfolding of a protein, self-assembly of polysaccharides in plant cell walls, enzymatic degradation of plastic, etc., are still beyond the state-of-the-art all-atom MD implementations. For such cases, coarse-grained (CG) methodologies were introduced to reduce the number of degrees-of-freedom, which has a direct impact on the number of simulated particles. In addition, they employ soft potentials which allow the use of larger time steps. In general, they are developed on the basis of rigorous statistical mechanics. Thus, they can deal with larger systems and longer time scales than all-atom MD. Today, one can find several CG models that have been employed to model large fluctuations of cellulose microfibrils. For example, Fan et al. [14] developed a one-site CG model for cellulose Iβ which described fibrils on a 10–500nm length scale. Srinivas et al. [15] presented a one-site CG model derived by the force-matching approach, which captures the amorphous state of cellulose Iβ and calculated the free energy of transition between the crystalline to amorphous state. Similarly, our [8] CG model with one-bead per D-glucose centered on C4 atoms was able to model cellulose I allomorphs. In addition, the popular Martini force field [16] has been employed to model crystalline native cellulose (i.e., Iβ). Other CG models for sugars [17,18,19] are capable of describing accurately complex sugars (e.g., DNA, RNA, etc.) and capture mechanical properties and dynamics, but they can not reach very large length scales, as required by cellulose fibrils (>40 nm). Some of these models rely on implicit water and thus they can not be used in aqueous conditions, therefore, failing to reproduce the experimental conditions necessary for industrial applications. However, a Martini description could be used in combination with other biomolecules in water such as proteins, lipids, nucleic acids, and other biopolymers. In this regard, a generic framework can be derived in terms of the Martini approach.

Here, we present a Martini 3 model for cellulose fibrils, which employs one set of potential parameters, able to differentiate three structures of cellulose, namely Iα, Iβ, and II. In order to describe the inner structure of the flat ribbon of a cellulose chain, we employed a set of harmonic constraints between nearest neighbours.

## 2. Materials and Methods

### 2.1. All-Atom MD Simulation

The initial structures of cellulose I allomorphs (Iα and Iβ) and cellulose II fibrils were generated by the Cellulose-Builder toolkit [20]. In practice, we prepared systems with 36 cellulose chains, each one composed of 100 D-glucose units. The final structures were visualized by VMD [21], and data were postprocessed using its internal protocols. All-atom MD simulations were performed using NAMD [22] version 2.14. The fibrils were modeled using the CHARMM36 force field [23], where the D-glucose denoted as the “BLGC” residue in the force field was used to describe each residue in a given cellulose chain. A triclinic box was used to represent the simulation box, and periodic boundary conditions were implemented in all directions. Solvation of the simulation box by TIP3P water molecules [24] included a buffer distance of 15 Å from the fibril. A total of 71,286, 67,403, and 59,383 water molecules were necessary for each fibrils system, namely Iα, Iβ, and type-II. The system had an average density of 0.1053 atoms.

The equilibration protocol was: (i) 10,000 of energy minimization via conjugate gradient protocol for the solvent, while the solute remained restrained; (ii) unrestrained MD simulation using a time step of 2 fs, and a reference temperature and pressure of 300 K and 1 atm, respectively. For the second step the Langevin thermostat and Piston, which are implemented in NAMD were employed. Long-range electrostatic interactions were computed using the PME [25] methodology; (iii) finally, a production run of 100 ns for each system was started after the target temperature and pressure were achieved in the NVT and NPT ensembles. Cross sections of the fibril structures are represented in Figure 1.

### 2.2. Coarse-Grained Model: Martini 3

In this part, we employed a versatile CG force field denoted as Martini 3 [27]. This new methodology has been successfully validated in several applications of large conformational changes in complex systems [28,29,30]. Hence, our aim was to build a robust CG model that not only reproduced the basic structural parameters of cellulose Iα, Iβ, and type-II fibrils but also a CG model, which can be transferable between different fibrils. In order to achieve this goal, we searched in the parameter space of all CG Martini bead types. Since each cellulose fibril has different interchain long-range interaction networks (i.e., different packing structures due to HBs) we had to select an elementary fragment of the cellulose chain that represents all the structures. This study used the whole cellulose chain as the basic building block (BB) for all different fibrils.

For this purpose, we modeled each cellulose fibril as an aggregate of the same BB. Each BB consisted of several planes, with each one defined by 4 CG beads denoted as CG1, CG2, CG3, and CG4 (see Figure 2). The orientation between planes changed along the fibril axis as a consequence of our parameterization. Validation of this minimal model was supported by all-atom MD data. There are several ways to parameterize a plane, and here, we used an elementary parametrization known as the simplicial complex homeomorphic (SCH) to a plane [31]. This mathematical construction is defined by triangular patches that optimally cover an entire surface, in our case the cellulose chains. CG simulations showed an average density of 0.0105 atoms/Å${}^{3}$, with an average of 51,291 atoms per box.

## 3. Results

### 3.1. Mapping CG Structure by All-Atom MD

Figure 3 shows atomistic distributions of distances between chosen CG sites after mapping via the SCH protocol. For all cellulose fibrils, we observe almost a well-defined bell-shape distribution with an exception for O2-O6 and O3-O6, where a bimodal distribution was sampled. In order to maintain a certain degree of parameter transferability between CG models of cellulose fibrils, we considered a distance value between two CG beads consistent with the average of the means of distributions. For those two cases where distributions diverged, we tuned the distance value to reproduce structural consistency in all CG models. The selected parameters are shown in Table 1.

### 3.2. Parameterization of the CG Model for Several Cellulose Fibrils

CG systems were simulated using GROMACS 2020.2 [32]. A similar equilibration procedure as described in the all-atom MD section was considered. In particular, long-range interactions were computed using Martini 3 standard methodology [27], namely, the reaction-field method [33] using a cutoff of 1.1 nm. Temperature and pressure control were achieved using the V-rescale thermostat [34] and Parrinello-Rahman barostat [35], respectively. Temperature and pressure were set to 300 K and 1 bar for all CG simulations. Equilibration in the NVT and NPT ensembles lasted 0.1 ns and 1 ns, respectively. The positions of two CG beads, namely CG1 and CG2, were constrained and fully relaxed during 1 ns by the end of the equilibration.

The time step for production MD runs was 0.02 ps and, CG-MD trajectories were set to 50 ns for each of the 18 different Martini 3 interaction types (i.e., P1, P6, SP1, SP6, TP1, TP6, N1, N6, SN1, SN6, TN1, TN6, C1, C6, SC1, SC6, TC1, and TC6). In the Martini 3 library, each interaction type is defined by at least three parameters, i.e., $\sigma_{ij}$ and $\epsilon_{ij}$ correspond to the Lennard-Jones (12-6) potential and q for particle charges [16]. In our case, we decided to parametrize the CG3 with a TC1 Martini 3 bead type as it has the smallest interaction with CG water, generally given by W bead type, and other CG beads. According to the all-atom MD distribution of distances between CG centers (see Figure 3), we defined the same elastic constant equal to 30,000 kJ/mol for well-behaved bell-shape cases. Only those flexible bond distances, “O2-O6” and “O3-O6”, required special attention. In this case, we tuned the elastic strength to a lower range of values using 25 kJ/mol, 250 kJ/mol, and 2500 kJ/mol. A cumulative time of about 3 μs was necessary to determine the best model amongst all parameters.

Figure 4 shows the results of the CG parametrization for several bead types in Martini 3 and the elastic bond strength for O2-O6 and O3-O6. The final set of parameters satisfy the criterion of a small deviation of the RMSD. This quantity was averaged over all the CG trajectory, using as a reference the last frame of AA-MD simulation. As a result we selected the bead type SP6 and 2500 kJ/mol for the elastic constant as they follow our structural criterion, and the RMSD remained smaller for all fibrils simultaneously (see Table 2).

### 3.3. Structural Validation of the Martini 3 Model for Cellulose Fibrils

Table 3 shows the comparison of distance in CG and MD simulations. The Martini 3 model with parameters SP6 bead type and 2500 kJ/mol for O2-O6 and O3-O6 intramolecular distances maintains accurately the internal structure of the cellulose chain in each cellulose fibril type, namely Iα, Iβ, and type-II. Hence, it highlights a transferable model between those three systems.

Our CG model preserves the internal structure of the D-glucose molecules and shows thermal stability in a range of temperature from 300 K–400 K (see Appendix A). These results indicate the possibility to study new applications in Martini 3 such as the structural relaxation of fibrils under different temperatures.

We also carried out a RMSD cross-correlation analysis. We computed an RMSD for each pair of frames from AA and CG trajectories and, subsequently, the RMSD average and respective standard deviations (SD) for each pair of trajectories. This combinatorial problem required the computation of 663,522 values of RMSD from a pair of structures. Note that this procedure captures the relative structural changes from two structures at different time points. Table 4 shows the RMSD cross-correlation for AA and CG trajectories. This result validates our Martini 3 model of several cellulose fibrils, as it shows the correct structural description for each crystalline cellulose fibril due to a small deviation in RMSD in the range of 5 Å. In fact it shows a well-equilibrated CG structure respect to its analogous AA system. Furthermore, the larger deviation with respect to other crystals (RMSD > 15 Å) shows that a given CG fibril model retains a structural distance from other cellulose crystals, as expected. Thus no transition or relaxation from one fibril to to another is observed in our simulations.

Figure 5 shows the analysis of one feature in cellulose fibrils: the twist angle. In our case, we define it by the dihedral angle formed by O2 and C6 centers from two subsequent D-glucose molecules. As a result, we observe qualitative agreement between AA-MD and CG-MD representations for cellulose I allomorphs (Iα and Iβ). However, our CG model is indeed more accurate for the case of cellulose type-II.

A detailed analysis of the distribution of twist angles shows consistency between AA and CG for type-II cellulose with a mean value of the twist angle about −0.42 degrees. In the case of cellulose I allomorphs, the dihedral angle in CG simulations reported about half of the value in AA-MD (see Table 5).

## 4. Conclusions

In this study, we employed a mathematical construction called geometrical simplicial complex for the determination of an optimal CG mapping scheme. In this context, this approach offers the advantage that a CG model does not require angular (three or four centers) potentials to be represented. However, a clear limitation such as the case of heterogeneity in complex polysaccharides may require a generalization of our approach.

We have tested our approach in two cellulose I allomorphs (i.e., Iα and Iβ), which are almost indistinguishable at large length scales. Although the local structure was mostly dominated by HBs, we managed to account for a versatile Martini 3 CG model capable of reproducing a single cellulose chain structure using only three uncharged Martini bead types. Furthermore, we also modelled the structure of cellulose type-II, which is substantially different from cellulose I allomorphs. Furthermore, our model captured the atomistic twist angle inherited by the fibril structure in solution for all three structures. Future experiments can address the relevance of the twist as a function of the temperature.

Our model opens the possibility to study large conformational changes of polysaccharides, as it is fully compatible with our earliest implementation of the Gō-Martini approach for the modelling of large conformational changes of proteins [36]. In this regard, a future improvement of this model for larger conformational changes inside a single CG cellulose chain can be achieved by switching some of the interactions between nearest neighbours by a consistent set of Gō-like potentials following standard rules for contact map creation [7,37]. Similar ideas, such as the dual-basin Gō-like model to capture transitions between different fibrils, can be suitable to speed up transitions [38,39].

## Figures and Tables

**Figure 1 molecules-27-00976-f001:**
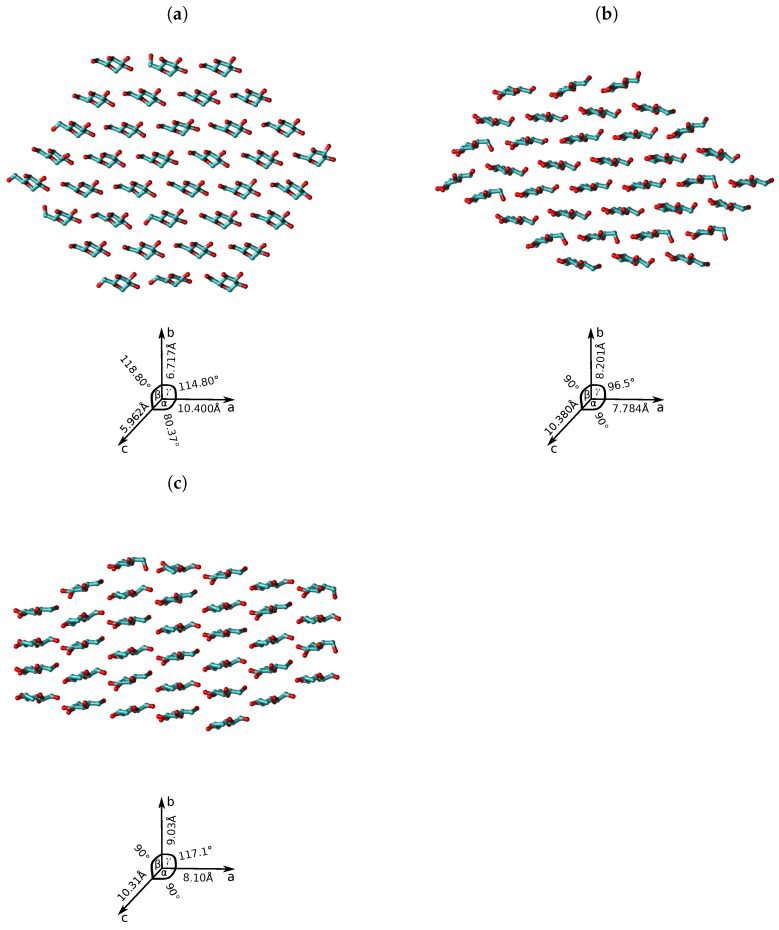
Snapshots of the cross-section for the relaxed atomistic structures of crystalline cellulose fibrils: (**a**) Iα, (**b**) Iβ, and (**c**) type-II, and and crystallographic unit cells are shown on the right side, and the values are taken from Refs. [9,10,26]. D-glucose molecule is shown in licorice representation with oxygens atoms in red and carbon atoms in cyan colours. Hydrogens were not included in this representation for sake of clarity. The chemical structure of cellulose fibrils were build by Cellulose-builder toolkit and rendered by VMD software.

**Figure 2 molecules-27-00976-f002:**
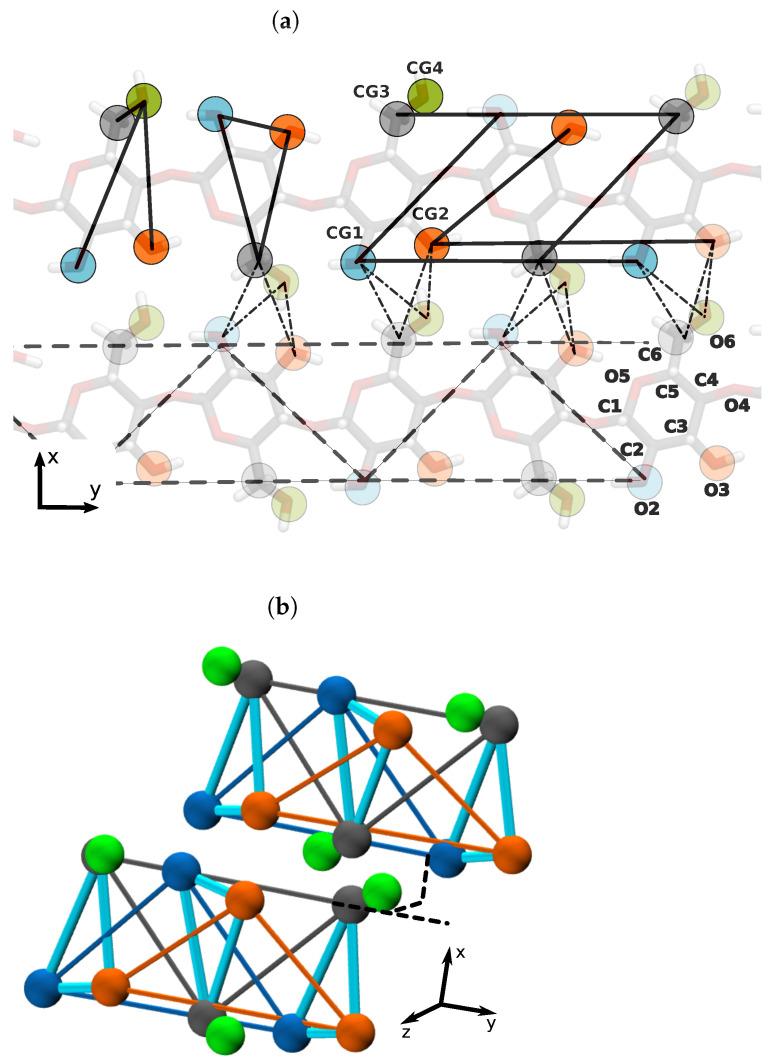
(**a**) Cellulose sheet representation of the CG model for the cellulose fibrils. Mapping of the atom, O2, O3, C6, and O6 into CG beads, CG1, CG2, CG3, and CG4, respectively, (top panel). Names of the heavy atoms in D-glucose are shown on the lower right. Solid lines indicate the bonded interactions of the model, dashed lines show the plane triangulation of a single cellulose chain, and thin dashed lines represent the non-bonded interaction between two parallel chains. (**b**) Equilibrated snapshot of two CG cellulose chains from two cellulose sheets for the Iα allomorph. The intra-residue bonds are depicted using cyan and inter-residue bonds by grey and orange.

**Figure 3 molecules-27-00976-f003:**
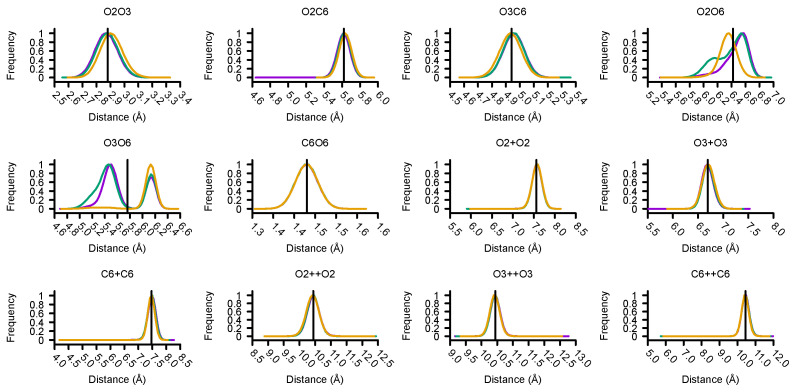
All-atom MD distribution of distances between atomic centers considered as CG sites. The labels assigned to distances are described in the method section, as well as the relationship between first and second D-glucose neighbours denoted by “+” and “++” respectively. Data for cellulose Iα, Iβ and type-II are represented by purple, green, and yellow solid lines, respectively. The horizontal black line indicates the value of the optimal distance parameter in the CG model. The distributions are bell-shaped with the exceptions of the intra-glucose distances O2-O6 and O3-O6, which follow a bimodal distribution.

**Figure 4 molecules-27-00976-f004:**
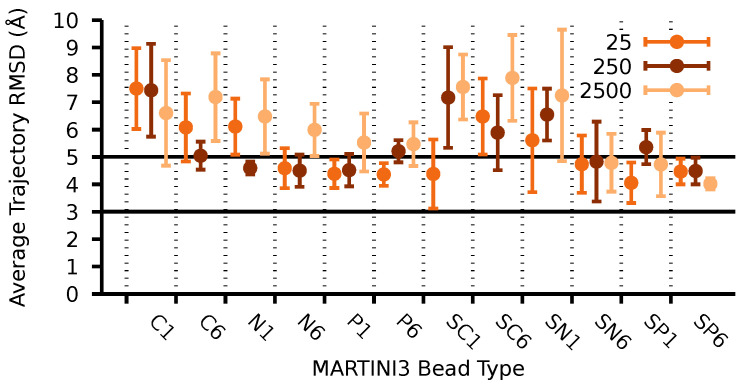
Time average RMSD of Iα, Iβ, and type-II cellulose fibrils as a function of the Martini 3 bead types and three different elastic constants for O2-O6 and O3-O6 intramolecular distances. Reference snapshot for each system is the last frame from all-atom MD simulations. Some bead types are not shown in the plot as they induced large CG-MD instability during the simulation.

**Figure 5 molecules-27-00976-f005:**
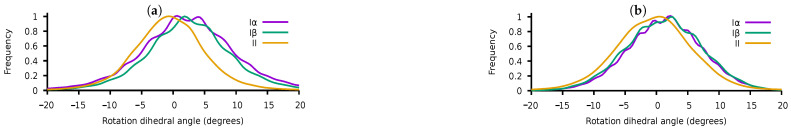
Distribution of the fibril twist angle defined by the dihedral angle formed by O2 and C6 centers from two subsequent D-glucose. Panel (**a**) shows AA-MD representation and bottom panel (**b**) for the CG-MD case.

**Table 1 molecules-27-00976-t001:** Optimal CG bonded parameters for cellulose I allomorphs and type-II fibrils. Parameters are defined within a D-glucose and between the two subsequent neighbours.

D-Glucose Neighbours	Center Pairs	$r_{ij}$ (Å)	$k_{ij}$ (kJ/mol)
none	O2-O3	2.88	30,000
none	O2-C6	5.62	30,000
none	O3-C6	4.94	30,000
none	C6-O6	1.43	30,000
none	O2-O6	6.42	2500
none	O3-O6	5.76	2500
1st	O2-O2	7.56	30,000
1st	O3-O3	6.69	30,000
1st	C6-C6	7.47	30,000
2nd	O2-O2	10.44	30,000
2nd	O3-O3	10.44	30,000
2nd	C6-C6	10.44	30,000

**Table 2 molecules-27-00976-t002:** Optimal Martini 3 nonbonded parameters for cellulose I allomorphs (Iα and Iβ) and type-II. CG3 is defined by bead type TC1 and CG1, CG2 and CG4 as SP6 bead type.

Interaction between CG Bead Types		$\sigma_{ij}$ (Å)	$\epsilon_{ij}$ (kJ/mol)
W	SP6	4.250	4.530
SP6	SP6	4.100	4.290
W	TC1	4.150	0.550
TC1	TC1	3.400	1.510
SP6	TC1	4.840	0.890

**Table 3 molecules-27-00976-t003:** Comparison of the distances between CG centres between AA-MD and CG-MD models for Iα, Iβ, and type II fibrils. Mean value and standard deviation are shown for each entry. The first column indicates whether the distance is found inside the D-glucose (none) or between the two subsequent (1st and 2nd neighbours) D-glucose molecules.

D-glucose Neighbours	Pairs	Fibril	Distances (Å)	
			AA-MD	CG-MD SP6/2500
None	O2-O3	Iα	2.87 ± 0.28	2.88 ± 0.23
None	O2-O3	Iβ	2.88 ± 0.28	2.88 ± 0.23
None	O2-O3	II	2.91 ± 0.28	2.88 ± 0.23
None	O2-C6	Iα	5.62 ± 0.27	5.63 ± 0.25
None	O2-C6	Iβ	5.63 ± 0.27	5.63 ± 0.25
None	O2-C6	II	5.64 ± 0.27	5.62 ± 0.25
None	O3-C6	Iα	4.96 ± 0.28	4.94 ± 0.25
None	O3-C6	Iβ	4.95 ± 0.28	4.93 ± 0.25
None	O3-C6	II	4.93 ± 0.28	4.94 ± 0.25
None	C6-O6	Iα	1.43 ± 0.16	1.43 ± 0.27
None	C6-O6	Iβ	1.43 ± 0.16	1.42 ± 0.27
None	C6-O6	II	1.43 ± 0.16	1.42 ± 0.27
None	O2-O6	Iα	6.47 ± 0.42	6.29 ± 0.57
None	O2-O6	Iβ	6.38 ± 0.46	6.31 ± 0.57
None	O2-O6	II	6.35 ± 0.35	6.57 ± 0.49
None	O3-O6	Iα	5.69 ± 0.58	5.65 ± 0.56
None	O3-O6	Iβ	5.62 ± 0.60	5.64 ± 0.56
None	O3-O6	II	6.08 ± 0.47	5.49 ± 0.50
1st	O2-O2	Iα	7.57 ± 0.35	7.56 ± 0.25
1st	O2-O2	Iβ	7.57 ± 0.34	7.56 ± 0.25
1st	O2-O2	II	7.58 ± 0.35	7.56 ± 0.25
1st	O3-O3	Iα	6.69 ± 0.34	6.70 ± 0.24
1st	O3-O3	Iβ	6.70 ± 0.34	6.71 ± 0.24
1st	O3-O3	II	6.70 ± 0.35	6.70 ± 0.24
1st	C6-C6	Iα	7.48 ± 0.38	7.46 ± 0.26
1st	C6-C6	Iβ	7.47 ± 0.37	7.47 ± 0.26
1st	C6-C6	II	7.46 ± 0.36	7.47 ± 0.26
2nd	O2-O2	Iα	10.44 ± 0.45	10.44 ± 0.24
2nd	O2-O2	Iβ	10.44 ± 0.44	10.43 ± 0.24
2nd	O2-O2	II	10.42 ± 0.45	10.44 ± 0.24
2nd	O3-O3	Iα	10.44 ± 0.40	10.43 ± 0.24
2nd	O3-O3	Iβ	10.44 ± 0.39	10.43 ± 0.24
2nd	O3-O3	II	10.42 ± 0.40	10.44 ± 0.24
2nd	C6-C6	Iα	10.44 ± 0.46	10.44 ± 0.26
2nd	C6-C6	Iβ	10.44 ± 0.46	10.44 ± 0.26
2nd	C6-C6	II	10.42 ± 0.45	10.44 ± 0.26

**Table 4 molecules-27-00976-t004:** RMSD cross-system for all cellulose fibril trajectories and their respective standard deviations.

AA-MD	CG-MD	Trajectory RMSD(Å)
Iα	Iα	4.264 ± 0.751
Iα	Iβ	16.860 ± 0.127
Iα	II	21.111 ± 0.235
Iβ	Iα	16.867 ± 0.120
Iβ	Iβ	5.440 ± 0.681
Iβ	II	18.786 ± 0.390
II	Iα	20.290 ± 0.091
II	Iβ	17.840 ± 0.130
II	II	3.740 ± 0.400

**Table 5 molecules-27-00976-t005:** Statistics of twist angles for all cellulose fibrils. The error in the measurements corresponds to the standard deviation (SD) for each entry.

	Average	SD	Max	Min
AA-MD				
Iα	3.557587	11.986272	−136.20	123.50
Iβ	2.550789	7.222558	−167.00	99.80
II	−0.420902	6.886370	−150.00	173.40
CG-MD				
Iα	1.406122	6.499510	−54.10	61.10
Iβ	1.251399	6.292622	−33.80	89.30
II	−0.427939	9.148864	−107.20	120.20

## Data Availability

The data presented in this study are available on request from the corresponding author. Source code is available in https://github.com/rams-research/cgcellulosefibril, accessed on 27 January 2022.

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
