# Peer review of "Martini 3 Model of Cellulose Microfibrils: On the Route to Capture Large Conformational Changes of Polysaccharides"

_molecules, 2022, doi:10.3390/molecules27030976_

Round 1

Reviewer 1 Report

  1. Whether is the water effect on the structure parameters of cellulose considered in the simulation? Especially on the hydrogen bond.
  2. It seems that the structure parameters of cellulose chains in Figure 1 are different from those of the classical Meyer-Misch cellulose crystallographic unit cell. Why?

Which molecular conformation of D-glucose (tg, gg, or gt) was used for the structure simulation of cellulose? Cellulose â…  and â…¡ are different.

  1. There should be a misalignment between different cellulose chains in the crystalline unit cells. However, in figure 2, the two chains are almost parallel to each other without position slip.
  2. Page 6, Line 151: Table 2?
  3. Page 8, Line 171: … may required …

required → require

Author Response

enclosed as a PDF file

Reviewer 2 Report

The Article presenting novel MARTINI 3 development is very interesting, however several control simulations proving quality of the model are missing.

  1. Long simulation presenting RMSD vs time of 3 three systems should be computed. Furthermore, each simulation system should have RMSD computed to other one. For example system starting type-II should have RMSD computed to Ialpha, Ibeta and type-II. To truly show stability of the simulated system.
  2.  Authors did not implement virtual valence bond angles. The behaviour of this degree of freedom during the simulation should be tested.
  3. Authors should justify the strange (as for MARTINI force field) mapping as not 4:1 heavy atoms are mapped.
  4. Simulation of stability of the system vs temperature should be tested.

Minor issues:

Better review of current coarse-grained methods enabling simulation of sugars should be performed.

Author Response

enclosed as a PDF file.

Round 2

Reviewer 2 Report

I am satisfied with answers. Authors did a good job of improving their article and removing my concerns. The article can be accepted in present form.

However, those results (in answers) should be added to Supplementary Information if possible. They are valuable and interesting results, proving that the model do not overstabilize the structure and that the system behavior in terms of angles is correct. Slight error is axes change (frequency should be angle and vice versa).

I am also pleased with the explanation to "Authors should justify the strange (as for MARTINI force field) mapping as not 4:1 heavy atoms are mapped" However it should be in main text or at least in supplementary information.